# Vegetation Dynamics in the Qinling-Daba Mountains through Climate Warming with Land-Use Policy

Yonghui Yao [1,*] and Lulu Cui [1,2]

1   State Key Laboratory of Resources and Environment Information System, Institute of Geographic Sciences and Natural Resources Research, CAS, Beijing 100101, China
2   University of Chinese Academy of Sciences, Beijing 100049, China
*   Correspondence: yaoyh@lreis.ac.cn

**Abstract:** The Qinling-Daba Mountains in central China (also known as the north–south transitional zone) comprise an ideal area to study land cover change, climate change, and human activities. The normalized difference vegetation index (NDVI) change and associated driving factors are highly sensitive to vegetation cover change. To discover the long-term vegetation trends in the transition zone and determine the driving factors of NDVI change in recent decades, this study analyzed the NDVI variation trend and its spatial variation with elevation, slope, and land-use type based on annual growing season NDVI data from 1990–2019 (Landsat 30 m; Google Earth Engine). The results show that NDVI values in the Qinling-Daba Mountains significantly increased and experienced a dynamic change process, involving an initial decrease and subsequent increase over this time period. The period of 2000–2005 showed a remarkable increasing stage of the NDVI in the transition zone. Such NDVI changes are sensitive to elevation and slope. For example, areas at elevations < 1500 m or with slopes of 5°–25° exhibited a stronger rate of NDVI increase than in other places. The NDVI change was also found to be positively affected by human land use and climate warming, both of which had a stronger impact than precipitation. The area with rapid NDVI growth was also the region with the greatest impact of human cropland and host to the Grain-for-Green project. This demonstrates that human land use has had a positive impact on the NDVI change in recent decades, although urbanization had led to a decrease in the NDVI in surrounding areas. Land-use policies have contributed to the large increase in NDVI values, especially those for forest conservation and expansion programs such as the Grain-for-Green project.

**Keywords:** north–south transitional zone of China; Qinling-Daba Mountains; normalized difference vegetation index; human activities; climate warming; Grain-for-Green Policy

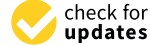



## 1. Introduction

Satellite data from China and India indicate that the Earth is greening in response to land-use management [1]. Studies have shown that vegetation change is strongly influenced by global climate change and human activities [1,2]. The normalized difference vegetation index (NDVI), which is correlated to the leaf area index [3] and gross primary production [4], has been widely used to analyze vegetation dynamics and its drivers as an indicator for vegetation coverage and biomass [5–11]. Previous studies of NDVI trends in China have shown that the NDVI has generally increased on national and regional scales [11–15]. Studies addressing the correlation between NDVI and climate have shown that the NDVI at the national scale is more sensitive to temperature than precipitation in North America and China and has shown an increasing trend corresponding to global warming [2,16,17]. In contrast, at the regional scale, the NDVI might be more sensitive to temperature or precipitation [2,11,13,18,19]. The impact of human activities on the NDVI is difficult to quantitatively analyze. Rapid urbanization has resulted in a sharp NDVI reduction in the Yangtze River and Pearl River deltas in China [20], whereas irrigation and

fertilization may have contributed to an NDVI increase in China and India [1,20], especially in areas of the National Natural Conservation Project and Grain-for-Green Project in China, which have effectively improved the regional vegetation coverage [1,13]. Climate change plays an important role in the NDVI trend patterns [20–22], and human activities (e.g., land-use management, urbanization) have exerted a large effect on the spatiotemporal patterns of NDVI trends in some regions [1,20]. Recent studies have disentangled the impacts of human activity and climate warming on vegetation dynamics at regional [23,24] and continental scales [25,26].

The Qinling-Daba Mountains, also called the north–south transitional zone of China, run through central China in an east–west direction and play an important role in China's geoecological pattern [27,28]. This range connects the Qinghai–Tibet Plateau and the North China Plain from east to west and separates the temperate and subtropical zones from north to south. The vegetation in this area, characterized by complexity, heterogeneity, and transition, is particularly sensitive to climate change [29,30]. Contemporary climate warming therefore requires a more in-depth understanding of vegetation change in this area.

Many studies have addressed vegetation change in the Qinling-Daba Mountains using the NDVI, and generally shown increased NDVI trends in this area [13,18,31–35]. Climate change, especially temperature and precipitation, has a great effect on vegetation. Luo et al. (2009) studied the relationship between the NDVI and climate in the Qinling-Daba Mountains using the GIMMS NDVI and SPOT NDVI datasets from 1982 to 2003, and reported that the correlation of NDVI and temperature was greater than that of NDVI and precipitation; the increased NDVI in the Qinling-Daba Mountains was an important response of global warming [18]. Zhang et al. (2011) studied the response of vegetation on Taibai Mountain to climate change and also showed that the NDVI was highly sensitive to temperature [19]. Using the SPOT VGT-NDVI ten-day dataset from 1998 to 2009, Ren et al. (2012) studied the NDVI trend in the Daba Mountains and its response to climate. Chen et al. (2019) analyzed the spatiotemporal change of vegetation in the Qinling-Daba Mountains using the GIMMS, SPOT VEG, and MODIS NDVI datasets, and found that the NDVI significantly increased from 1982 to 2017 and vegetation cover was positively correlated with temperature but not with precipitation [28]. They pointed out that the NDVI was significantly correlated with temperature and insignificantly correlated with precipitation [18,19,28,32]. However, other studies have shown opposing results; for example, Liu et al. (2015) stated that the change in vegetation coverage from 2000 to 2014 was mainly attributed to the deficit of precipitation [11]. Human activities also have a strong impact on vegetation change in the Qinling-Daba Mountains. Cui et al. (2012) showed that the temporal stability of vegetation was inversely distributed with distance from human aggregation areas [36]. Deng et al. (2018) argued that some human activities, such as the Grain-for-Green Project, promoted the NDVI increase, while others, such as mining and urbanization, led to a decrease in the NDVI [13]. Generally, human activities induced ambiguous effects on vegetation coverage in the Qinling-Daba Mountains: both positive effects (through the implementation of the ecological restoration project) and negative effects (through urbanization) were observed [11].

Although many studies have focused on the NDVI trends in the Qinling-Daba Mountains, most study areas were concentrated in Shaanxi Province [11,13,36,37] and did not cover the entire Qinling-Daba Mountain range. Those results could therefore not reveal the overall condition of the NDVI in this region. Data used in those studies were typically MODIS NDVI data [11,34,35] with a slightly lower spatial resolution for detailed spatial information. To bridge this gap, the aim of this study is to investigate the vegetation dynamics and influence of climate and human activities in the Qinling-Daba Mountains using the newest and highest-spatial-resolution NDVI data (Landsat NDVI data with 30-m resolution). For a deeper understanding of the vegetation variation in past decades, the time range in this study is confined to the growing season from 1990 to 2019. This work is motivated by the following questions: (1) What are the long-term vegetation trends in the transitional zone and have these trends significantly changed within recent decades?

(2) How do climate and human activities influence the NDVI variation, and what are the dominant factors in this region?

## 2. Study Area

The Qinling-Daba Mountains are situated in central China (102°–114° E, 30°–36° N) and are mainly composed of the Qinling Mountains, Hanzhong Basin-Hanshui Valley, and Daba Mountain, as well as the low hills in western Henan Province, low and middle mountains in northwestern Hubei Province, and parts of Sichuan and Gansu (Figure 1). This area spans two climatic zones, a warm temperate zone and a north subtropical zone, and the climate is warm and humid [38]. The annual average temperature in this region ranges between 12 °C and 16 °C, and the annual precipitation is approximately 400–1500 mm [39]. The vegetation in this region gradually transitions from subtropical evergreen broadleaved forest to deciduous broadleaved forest from south to north, and has vertical zonality in the mountains [27]. The subtropical evergreen broadleaved forest mainly includes communities dominated by *Fagaceae*, such as *Lithocarpus* spp., *Cyclobalanopsis glauca* (Thunb.) Oerst., and *Castanopsis fargesii* Franch. The main types of deciduous broadleaved forests are represented by various coenosis characterized by the dominance of *Quercus variabilis* Blume, *Q. acutissima* Carruth., *Q. liaotugensis* Koidz., *Populus davidiana* Dode, and *Betula platyphylla* Sukaczev. The main subtropical coniferous forests are pine forests dominated by *Pinus massoniana* Lamb. or *P. armandii* Franch., and spruce forests with *Picea asperata* Mast. Moreover, scrublands and thicket are quite frequent and are characterized by various species, such as *Rhododendron* spp., *Potentilla fruticosa* L., *Spiraea* spp., and *Phellodendron amurense* Rupr. [40]. This region marks the transitional zone of China and the vegetation exhibits complexity, biodiversity, and climate sensitivity, which is of great significance for China's geographical pattern, the evolution of biological flora, and the distribution of natural resources [27].

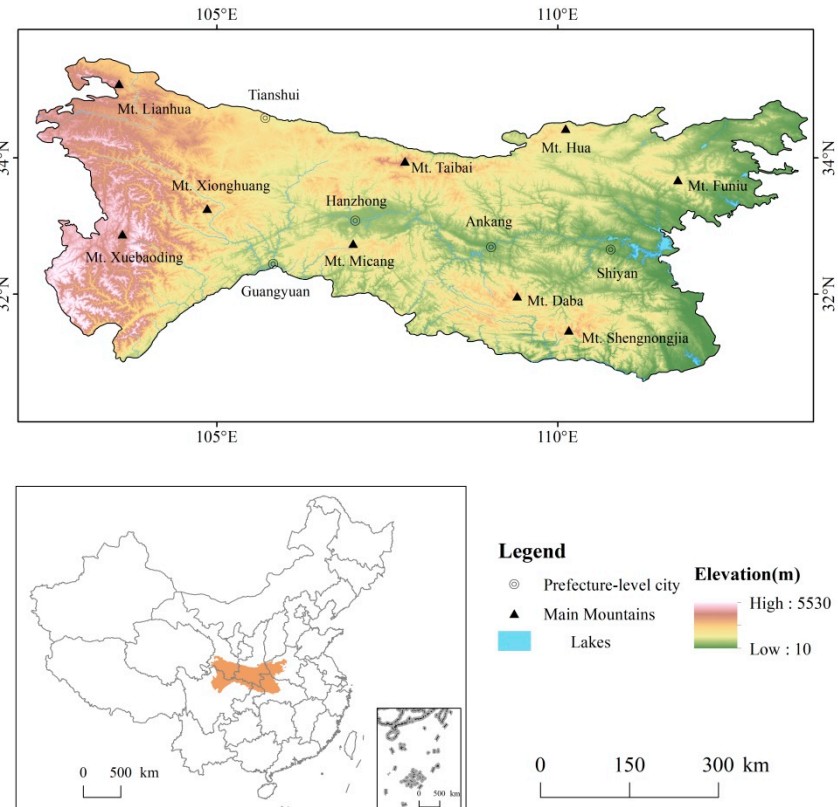

**Figure 1.** Location and landforms of the Qinling-Daba Mountains in China.

## 3. Datasets and Methods

### 3.1. Data

The NDVI dataset used in this study consists of the annual growing season NDVI (May to September) data (30 m resolution) of Landsat 5 from 1990 to 2011, Landsat 7 in 2012, and Landsat 8 from 2013 to 2019, which were synthesized using the maximum synthesis method on the Google Earth Engine (GEE) platform. Savitzky–Golay (SG) filtering was performed on the annual growing season NDVI data to further reduce the noise. Strip loss of the Landsat 7 ETM+ data in 2012 was resolved by interpolating between the 2011 and 2013 satellite data using the local window effective data. Roy et al. (2016) studied the difference between Landsat 8 OLI data and Landsat 5 TM and Landsat 7 ETM+ data, and proposed a fitting coefficient of mutual conversion based on ordinary least-squares regression (ETM+ = −0.0110 + 0.9690 OLI) [41]. This proposed fitting coefficient was used in this study to fit the Landsat 8 OLI data to the Landsat 7 ETM+ data to construct the NDVI data from the long-time series across the Landsat TM, ETM+, and OLI data in the study area.

Observed temperature and precipitation meteorological data (94 stations) were downloaded from the Data Center of Resources and Environment Science, Chinese Academy of Sciences (http://www.resdc.cn, accessed on 1 July 2019) for 1986–2018. The world climate data included temperature and precipitation data with a spatial resolution of approximately 1 km$^2$ (downloaded from http://www.worldclim.org/, accessed on 31 December 2017) and were generated by global meteorological station data using the Auspline interpolation method, including monthly temperature, monthly precipitation, annual average temperature, and annual precipitation data [42].

Land cover datasets in 1999, 2009, and 2019 were downloaded from http://irsip.whu.edu.cn/resources/CLCD.php (accessed on 30 June 2021), including the Landsat-derived annual China Land Cover Dataset (CLCD, 30-m resolution) on the GEE platform with annual land cover and dynamics in China from 1990 to 2019 [43]. These were used to analyze the influence of land-use type on NDVI. The SRTM DEM data derived from the Consortium for Spatial Information (CGIAR-CSI) with a 90 m spatial resolution were mainly used to analyze the trend of NDVI with variable altitude and slope.

### 3.2. Methods

The Sen trend method [44] was used to analyze the NDVI trend for 1990–2019. This approach effectively avoids the influence of time-series data loss and data distribution form, and eliminates the interference of time-series outliers [45]. The Mann–Kendall (MK) significant test and MK mutation test [46–48] were conducted to test the significance of the calculated Sen trend and determine the mutated NDVI change period. The temporal and spatial distributions of the NDVI were analyzed in five-year intervals based on the NDVI data and NDVI Sen trend for 1990–2019, and the increasing rates (the slopes of the trend lines) in the different elevations and slopes were calculated by linear trend analysis. To identify the driving factors of NDVI change, correlation analysis was applied to analyze the correlation between the NDVI and temperature/precipitation. The NDVI changes in land-use types were calculated to reveal the impact of human activities on the NDVI, and the impact of forest conservation and expansion programs in China on NDVI was discussed.

## 4. Results

### 4.1. Significant NDVI Increase in the Qinling-Daba Mountains

The results of the Sen trend analysis showed that the NDVI in 1990–2019 increased in most of the Qinling-Daba Mountains (>75% of the study area), decreased in only a few areas (3.5% of the study area), and remained relatively stable with no significant change in 20.7% of the study area (Figure 2). The increasing rate of NDVI was 0.3%/a ($R^2$ = 0.97) (Figure 2, left). The area with increasing NDVI was mainly located in the south and east of the Qinling-Daba Mountains and in places between Longnan and Tianshui to the west of the Qinling-Daba Mountains. The NDVI values in Longnan to Tianshui showed a significant

increase in the MK test ($p < 0.01$). The areas with both decreasing and insignificantly changing NDVI were clustered in the west and north of the Qinling-Daba Mountains, except for the area from Longnan to Tianshui. The NDVI also showed a significant downward trend around some cities such as Hanzhong City and Shiyan City. Another result was that there was no obvious change in NDVI in traditional nature reserves such as Shennongjia Nature Reserve, Taibai Mountain Nature Reserve, or subalpine and alpine areas such as the Daba Mountains and the west mountains of this region (Figure 2, left).

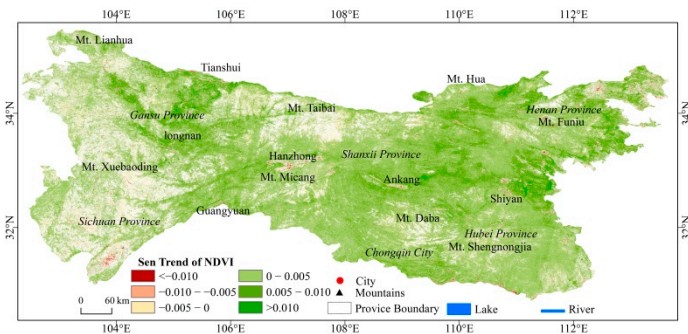 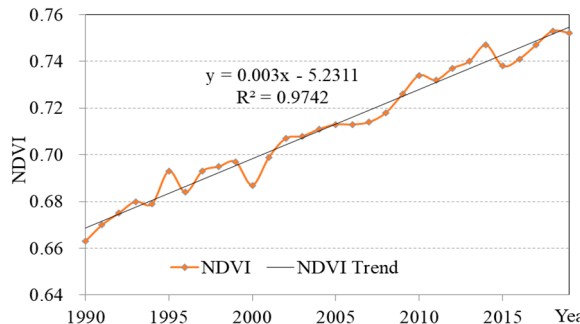

**Figure 2.** Map of the Sen trends in the annual growing season from the Landsat NDVI values for 1990–2019. (**Left**): Statistically significant trends (MK test, $p < 0.05$) are color-coded. White areas show the vegetated land with statistically insignificant trends ($p \geq 0.05$). Blue areas represent water. (**Right**): Trend of the annual average NDVI.

### 4.2. Temporal and Spatial Differences of NDVI Changes in the Qinling-Daba Mountains

The NDVI trend showed significant temporal and spatial differences in the Qinling-Daba Mountains (Figure 3). The results showed a significant downward trend for 1990–1994 and 1995–1999, but the western area of the Qinling-Daba Mountains initially increased in 1990–1994 and then decreased in 1995–1999, while the eastern area showed the opposite: an initial decrease in 1990–1994 followed by an increase in 1995–1999 (Figure 3a,b). The periods of 2000–2004 and 2005–2009 showed the fastest NDVI growth in most of the study area, except in the southwest and east (Figure 3c,d). The NDVI growth slowed after 2010 and even showed a downward trend in the middle of the Qinling-Daba Mountains. The NDVI values in the west of the Qinling-Daba Mountains continued to increase in 2010–2014 and 2015–2019, and those in the east showed either an increase or insignificant change (Figure 3e,f).

The MK mutation test result verified the increasing NDVI trend for 1990–2019 (MK test $p < 0.05$ after 1995) and a remarkable NDVI increasing time was found to occur in 2005 (Figure 3g). In each five-year dataset, 49%–64% of the area showed no significant change of NDVI; 25%–40% of the area showed increasing NDVI, reaching a maximum in 2000–2004 and 2005–2009; and 7%–20% of the area showed a decreasing NDVI trend (Figure 3h).

Spatially, the coefficients of NDVI variation showed a substantial change, mostly increasing, in the west and northeast of the Qinling-Daba Mountains and in the water-source conservation area of the South-to-North Water Transfer Project, whereas those of the traditional natural reserves (e.g., Shennongjia National Nature Reserve, Taibai Mountain National Nature Reserve) showed little change over the past 30 years (Figures 1 and 4).

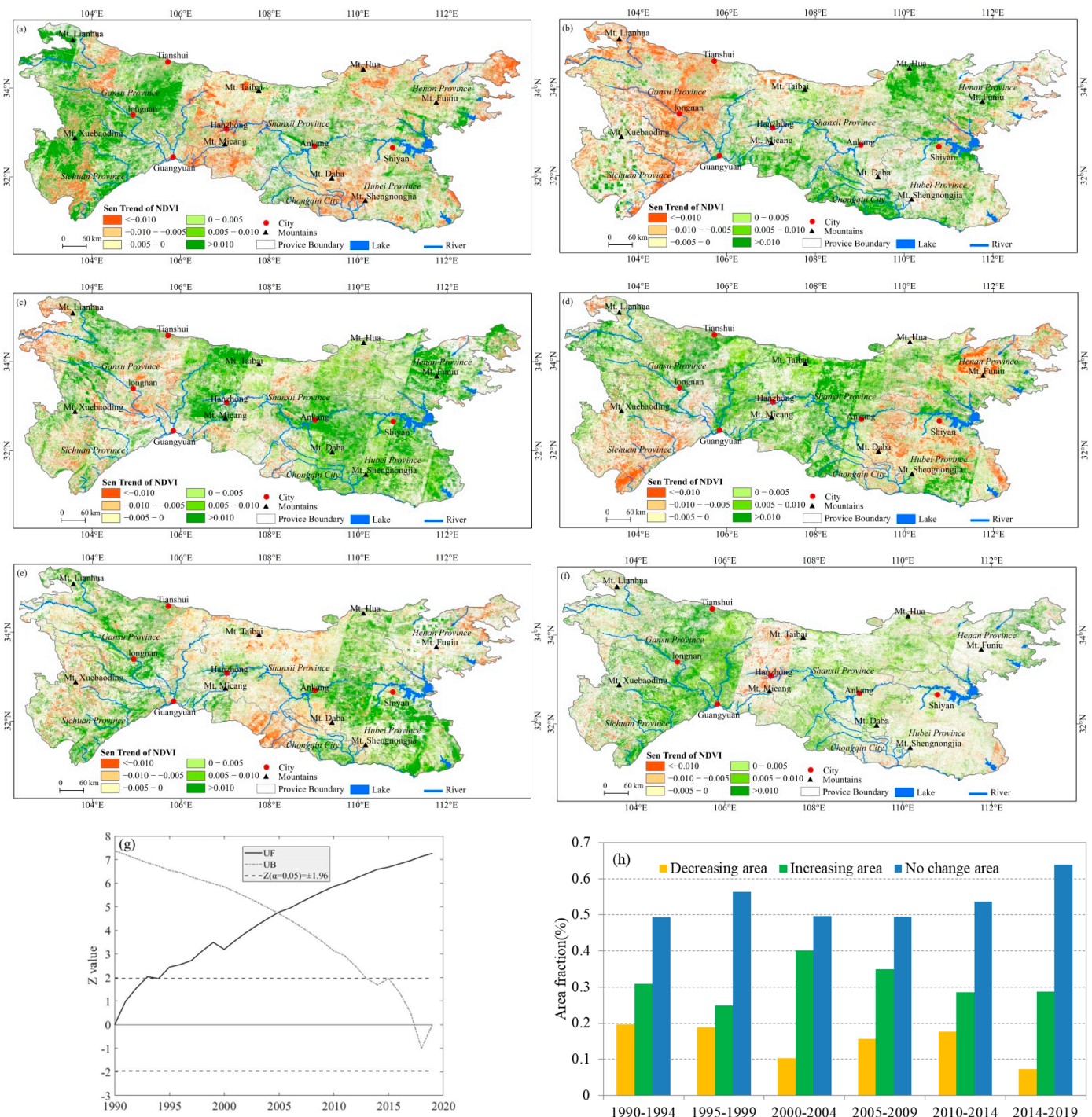

**Figure 3.** Sen trend maps for the annual growing season from Landsat NDVI values in different periods: (**a**) 1990–1994; (**b**) 1995–1999; (**c**) 2000–2004; (**d**) 2005–2009; (**e**) 2010–2014; and (**f**) 2014–2019. Statistically significant trends in the six periods (MK test, $p < 0.05$) are color-coded. White areas show the vegetated land with statistically insignificant trends ($p \geq 0.05$), and blue areas represent water. (**g**) MK mutation test of the NDVI change. (**h**) Areal fraction of the NDVI variation.

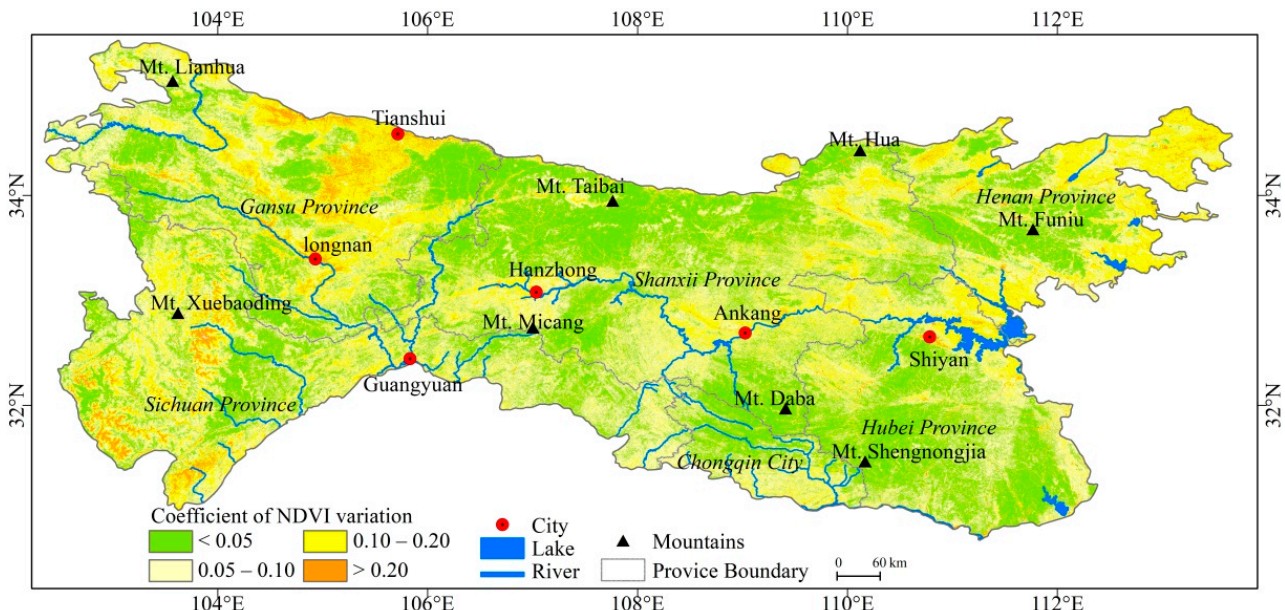

**Figure 4.** Coefficient of NDVI variations over the past 30 years. Coefficient values larger than 0.2 indicate large NDVI changes; values lower than 0.05 indicate minor NDVI changes.

### 4.3. Influence of Elevation and Slope on NDVI Changes

In 1990–2019, the increasing rate of NDVI decreased with ascending altitude, and was higher in areas at elevations < 1500 m compared with other altitudes. The highest rate was in the area with elevation < 1000 m and the lowest rate was in the area with elevation > 3000 m (Table 1). Similarly, the increasing NDVI rate also decreased with increasing slope, except for the areas with slopes shallower than 5°. The rate was higher in areas with slopes shallower than 5° compared with areas with slopes steeper than 35°, and the places with slopes of 5°–25° showed the fastest NDVI growth. (Table 1). This showed that the NDVI increase occurred faster in areas of low elevation (<1500 m) or shallow slopes (<25°).

Areas at elevations lower than 1000–1500 m in the Qinling-Daba Mountains are typically characterized by experiencing the strongest impact of human activities, and the impact weakens with increasing elevation. Flat areas (<15°–25°) in the Qinling-Daba Mountains are often cultivated by cropland and most strongly affected by human activities. The area of rapid NDVI growth therefore coincides with the area of greatest impact of human activities.

**Table 1.** Time-series variation of the mean NDVI values in areas of different elevation and slope for 1990–2019. The increasing rate of the mean NDVI decreased with increasing elevation, and was faster in areas at elevations < 1500 m compared with other elevations. The increasing rate of the mean NDVI also decreased with increasing slope angle, except for areas with slopes shallower than 5°.

| Year | Mean NDVI at Different Altitudes | | | | | | Mean NDVI for Different Slopes | | | | |
|---|---|---|---|---|---|---|---|---|---|---|---|
| | 0–1000 m | 1000–1500 m | 1500–2000 m | 2000–2500 m | 2500–3000 m | >3000 m | 0°–5° | 5°–15° | 15°–25° | 25°–35° | >35° |
| 1990 | 0.632 | 0.699 | 0.705 | 0.691 | 0.684 | 0.608 | 0.545 | 0.614 | 0.665 | 0.694 | 0.701 |
| 1995 | 0.654 | 0.724 | 0.734 | 0.722 | 0.725 | 0.674 | 0.571 | 0.646 | 0.695 | 0.726 | 0.733 |
| 2000 | 0.655 | 0.729 | 0.723 | 0.698 | 0.702 | 0.649 | 0.557 | 0.639 | 0.691 | 0.720 | 0.725 |
| 2005 | 0.693 | 0.752 | 0.740 | 0.716 | 0.719 | 0.657 | 0.588 | 0.669 | 0.718 | 0.743 | 0.745 |
| 2010 | 0.718 | 0.774 | 0.766 | 0.742 | 0.736 | 0.664 | 0.609 | 0.698 | 0.743 | 0.762 | 0.757 |
| 2015 | 0.727 | 0.778 | 0.762 | 0.743 | 0.737 | 0.662 | 0.613 | 0.703 | 0.749 | 0.763 | 0.759 |
| 2019 | 0.736 | 0.794 | 0.785 | 0.762 | 0.748 | 0.672 | 0.612 | 0.716 | 0.763 | 0.778 | 0.774 |
| Linear trend analysis | $Y = 0.0186x + 0.6134$ ($R^2 = 0.96$) | $Y = 0.0156x + 0.6874$ ($R^2 = 0.98$) | $Y = 0.0121x + 0.6966$ ($R^2 = 0.90$) | $Y = 0.0107x + 0.6821$ ($R^2 = 0.81$) | $Y = 0.0089x + 0.6859$ ($R^2 = 0.75$) | $Y = 0.0065x + 0.6290$ ($R^2 = 0.39$) | $Y = 0.012x + 0.5369$ ($R^2=0.87$) | $Y = 0.0171x + 0.6009$ ($R^2 = 0.95$) | $Y = 0.0162x + 0.6529$ ($R^2 = 0.96$) | $Y = 0.0131x + 0.6883$ ($R^2 = 0.94$) | $Y = 0.0108x + 0.6987$ ($R^2 = 0.91$) |

### 4.4. Climate Warming as a Driving Factor of NDVI Change

The results of the climate change analysis based on 94 meteorological observed stations for 1986–2018 showed that the climate of the Qinling-Daba Mountains was significantly warming (Figure 5a,b), while the precipitation slightly increased (Figure 5c,d). The temperature warming of 82 stations passed the 0.05 significance level test, among which 73 stations passed the 0.01 significance level test. The temperature warming rates were mostly between 0.2 and 0.5 °C/10a. The precipitation at each station also slightly increased. Ninety stations did not pass the 0.05 significance level test and the increasing precipitation rate was between 20 and 50 mm/10a (Figure 5c,d).

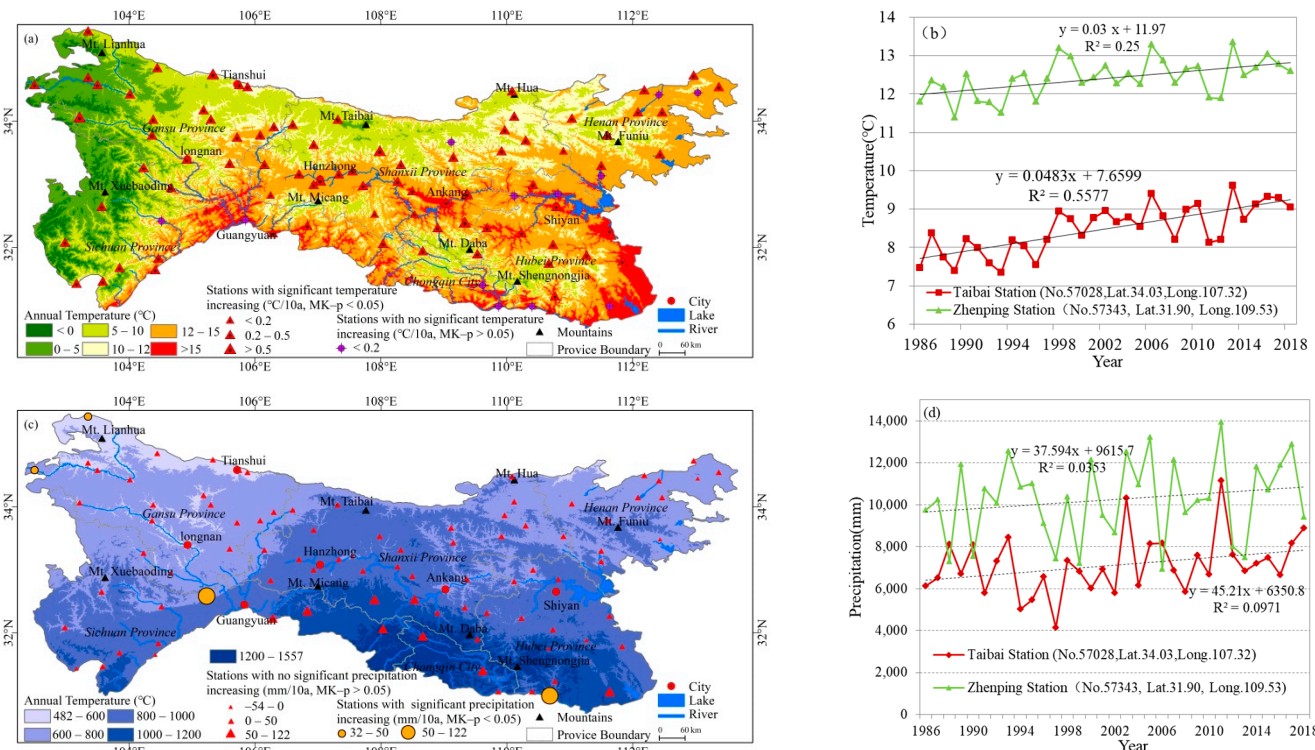

**Figure 5.** Spatial pattern of annual temperature and precipitation, and rates of temperature warming and precipitation obtained from meteorological observation stations over the past 30 years. (**a**) The temperature warming rates for 82 of 94 stations were >0.2 °C/10a and passed the significance test (MK test $p < 0.05$); only 12 stations had lower temperature warming rates and failed to pass the significance test (MK test $p \geq 0.05$). (**b**) Annual average temperatures of the Taibai Mountain station in the northern area of the Qinling-Daba Mountains and the Zhenping station in the southern area. Both showed significantly increasing trends with a greater increase rate of the former than that of the latter. (**c**) The precipitation increasing rate: only four stations passed the significance test (MK test $p < 0.05$), and most with an increasing rate between 20 and 50 mm/10a. (**d**) Annual precipitation showed no significant increase at the Taibai Mountain and Zhenping stations.

The correlation analysis results showed that temperature had both positive and negative effects on the NDVI. Thirty-seven stations had negative correlation coefficients with the NDVI, among which 16 stations (correlation coefficients < −0.35) passed the 0.05 significance level test; and 57 stations had positive correlation coefficients, among which 23 stations (correlation coefficients > 0.35) passed the 0.05 significance level test. Precipitation also demonstrated both positive and negative effects on the NDVI, but only four stations passed the 0.05 significance level test (Figure 5c). This showed that climate warming, rather than precipitation increase, has had an important impact on the vegetation cover change in the past three decades.

*4.5. Human Land Use as an Important Driving Factor of NDVI Change*

Vegetated lands cover more than 98% of the Qinling-Daba Mountains and include forests, croplands, grasslands, and shrub-woody lands. Among them, forests, croplands, and grasslands have a significant impact on the NDVI (Table 2). The forest coverage in the Qinling-Daba Mountains increased from 62.85% in 1990 to 69.16% in 2019, while the croplands, grasslands, and shrub-wood lands decreased from 19.71%, 14.97%, and 1.47% in 1990 to 16.82%, 11.65%, and 0.43% in 2019, respectively. Over the last three decades, 74.08% of the NDVI values have increased, 3.46% have decreased, and 20.09% have shown insignificant change in the four above land cover types (Table 2). Nearly three-fourths of the NDVI increase occurred in forests, and one-fifth was from croplands. Areas that showed an insignificant change of NDVI were also mainly from forests and croplands. These results demonstrated that human land use is an important driver of NDVI change in the Qinling-Daba Mountains.

**Table 2.** Proportion of NDVI change in the vegetated lands and impervious surface lands for 1990–2019, and the proportion of main land-use types of the Qinling-Daba Mountains in 1990 and 2019.

| Land-Use Type | Proportion of NDVI Change of the Qinling-Daba Mountains for 1990–2019 (%) | | | Proportion of Land Use Type of the Qinling-Daba Mountains (%) | |
|---|---|---|---|---|---|
| | Increasing | Decreasing | Insignificant Change | 2019 | 1990 |
| Forests | 54.47 | 1.65 | 13.04 | 69.16 | 62.85 |
| Croplands | 12.60 | 0.91 | 3.31 | 16.82 | 19.71 |
| Grasslands | 7.01 | 0.90 | 3.74 | 11.65 | 14.97 |
| Shrub-wood lands | 0.26 | 0.02 | 0.15 | 0.43 | 1.47 |
| Total for above four vegetated lands | 74.08 | 3.46 | 20.09 | 97.63 | 99 |
| Impervious surface land | 0.46 | 0.40 | 0.28 | 1.14 | 0.50 |

Impervious surface land area increased in the Qinling-Daba Mountains over the study period (1990–2019), and the proportion of NDVI in impervious surfaces also increased (Table 2). The NDVI in the surrounding areas of most cities and towns showed a significant downward trend or no significant change. For example, the NDVI around cities in the Hanshui River Basin significantly decreased or showed no significant change, whereas the NDVI in areas far from cities and towns showed a significant increase (Figures 2 and 6). This implied that urbanization had a negative impact on vegetation cover, which was consistent with previous studies [1,13].

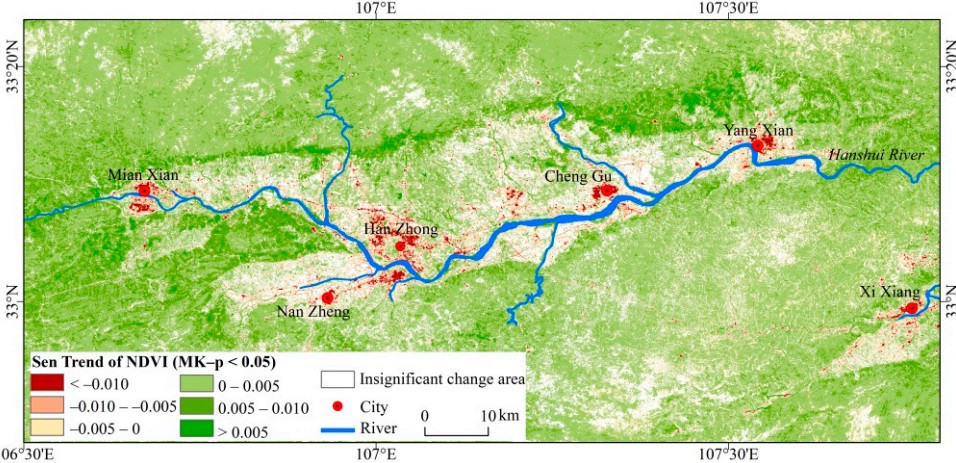

**Figure 6.** The NDVI Sen trend in the Hanshui River Basin. The NDVI in the areas surrounding cities and towns showed a significant downward trend or no significant change.

## 5. Discussion and Conclusions

### 5.1. Discussion

(1) Impacts of precipitation change on vegetation dynamics

Climate change, especially meteorological factors of temperature and precipitation, has a great effect on vegetation [49]. Our results showed that the climate warming played more important roles on vegetation dynamics in the Qinling-Daba Mountains than precipitation, which coincided with most of studies about the vegetation dynamics in this region [18,19,31–35]. However, there are some studies that show that the change in vegetation coverage was mainly attributed to the deficit of precipitation [11]. According to the annual precipitation of meteorological stations in the Qinling-Daba Mountains from 1986 to 2019 (Figure 5c,d), there has been no significant change in precipitation, though it has also generally gently increased over the past 30 years. In addition, annual precipitation in the Qinling-Daba Mountains ranges between 1500 mm in the south and 400 mm in the north (Figure 5c); there is sufficiently abundant precipitation for vegetation growth and development at all times. Thus, we think vegetation dynamics in the past three decades in this region is unlikely to mainly attribute to the deficit of precipitation or precipitation increase and is possibly a response of climate warming. One possible reason for the different results regarding the relationship between NDVI and precipitation might be the different resolutions of NDVI data used in these studies. Another possible reason might be that the analysis on a single time scale cannot accurately reflect the response mechanism of vegetation to climate change. Therefore, it is meaningful to study the relationship at multiple time scales, which contain annual, interannual, and interdecadal scales and long-term trends [49].

(2) Importance of human land use on vegetation dynamics

A recent study has indicated that the "Greening Earth" is attributed to human land-use practices in China and India [1]. In recent decades, China has implemented ambitious engineering programs to conserve and expand forests with the goal of mitigating land degradation, air pollution, and climate change, which plays an important positive role for vegetation cover changes [1]. The Qinling-Daba Mountains is located in central China and is a typical Chinese representative of the relation between land-cover change, climate change, and human activities [27]. Its vegetation cover change and driving factors also well reflect vegetation cover change in China. Our results showed that the significant NDVI increase occurred mostly in forest land, cropland, and grassland (Table 2), and the NDVI initially decreased in 1990–1999 and then increased after 2000, even with a remarkable increase in 2005 (MK mutation test, Figure 3g). This vegetation dynamic process is also closely related to China's land-use policy. After China implemented the land contract policy in 1980s, there was a large number of deforestation and reclamation events that caused severe soil erosion [50–52]. Upon recognizing the seriousness of this problem, the Chinese government issued the Grain-for-Green Policy in 1999–2000, toward which local governments formulated strict implementation measures [53,54]. For example, croplands with a slope steeper than 25° were required to be returned to forest (or grassland), and croplands with slopes between 15° and 25° were conditionally returned to forest or grassland. This explains why places with slopes between 15° and 25° have larger NDVI growth and faster NDVI increases after 2000–2005 (Table 1). Of course, more detailed information about the reforestation of Grain-for-Green needs to be acquired, and its effect on vegetation dynamics evaluated.

Moreover, areas at elevations lower than 1500 m (especially <1000 m) show the strongest impact of human activities on the areas with higher NDVI increase rates than at other elevations (Table 1). In addition to the reforestation of Grain-for-Green in the lower-elevation areas, cropland also contributed to the NDVI increase owing to the rapidly growing hybrid cultivars, multiple croppings, irrigation, fertilizer use, pest control, improved seed quality, farm mechanization, credit availability, and crop insurance programs [1], all of which demonstrate that human land use in the Qinling-Daba Mountains has recently played a positive impact on vegetation dynamics.

*5.2. Conclusions*

The NDVI change of the Qinling-Daba Mountains experienced a dynamic progression of an initial decrease followed by an increase, although it significantly increased overall from 1990 to 2019. The period of 2000–2005 was a period of remarkable NDVI change from decreasing to increasing. Meanwhile, the NDVI changes were also sensitive to the elevation and slope: areas at elevations < 1500 m or with a slope of 5°–25° showed a larger NDVI increasing rate than in other areas. Moreover, the NDVI change was found to be positively affected by human land use and climate warming. The area with rapid NDVI growth coincided with the area of the strongest impact of human cropland and the Grain-for-Green project, which supports the claim that that human land use has had a positive impact on the NDVI change in recent decades, although this urbanization led to an NDVI decrease in suburban areas. Land-use policies, especially those concerning forest conservation and expansion programs, such as the Grain-for-Green project, strongly contributed to the NDVI increase. The clear correlation between NDVI and temperature showed that the increased NDVI in the Qinling-Daba Mountains was also an important response of climate warming. The NDVI change of the Qinling-Daba Mountains and its driving factors are a representative example of vegetation cover change in China.

**Author Contributions:** Y.Y. designed the study, did the analysis and wroted the manuscript; L.C. collected the data. All authors have read and agreed to the published version of the manuscript.

**Funding:** This study was funded by the National Natural Science Foundation of China (Grant No. 41871350) and the Scientific and Technological Basic Resources Survey Program "Integrated Scientific Investigation of the North-South Transitional Zone of China" (Grant No. 2017FY100900).

**Informed Consent Statement:** This study did not involve humans.

**Acknowledgments:** The authors wish to thank Kou Zhixiang from the Institute of Geographic Sciences and Natural Resources Research, Chinese Academy of Sciences for processing the Landsat NDVI data. Our appreciation also goes to two anonymous reviewers, whose comments and suggestions helped to greatly improve the manuscript.

**Conflicts of Interest:** The authors declare no conflict of interest.

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
