# Peer review of "Vegetation Dynamics in the Qinling-Daba Mountains through Climate Warming with Land-Use Policy"

_forests, doi:10.3390/f13091361_

Round 1

Reviewer 1 Report (Previous Reviewer 2)

I suggest rewriting the vegetation description in 2. Study area as follows:   The subtropical evergreen broad-leaved forest mainly includes communities dominated by Fagaceae, such as Lithocarpus sp.pl., Cyclobalanopsis glauca (Thunb.) Oerst., and Castanopsis fargesii Franch. The main types of deciduous broad-leaved forests are represented by various coenosis characterized by the dominance of Quercus variabilis Blume, Q. acutissima Carruth., Q. liaotugensis Koidz., Populus davidiana Dode, and Betula platyphylla Sukaczev. The main subtropical coniferous forests are pine forests dominated by Pinus massoniana Lamb. or P. armandii Franch. and spruce forests with Picea asperata Mast. Besides, scrublands and thicket are quite frequent and are characterized by various species, as Rhododendron sp. pl., Potentilla fruticosa L., Spiraea sp. pl. and Phellodendron amurense Rupr.

Author Response

We are very appreciated this suggestion of the reviewer and accepted the suggestion. The rewriting paragraph is very professional and we can’t write such good sentences. Thanks a lot.

Reviewer 2 Report (Previous Reviewer 1)

The revision has addressed my comments and concerns. 

Author Response

We are very grateful to reviewers for their comments and suggestions. With their helps, the quality of this manuscript has been greatly improved. 

This manuscript is a resubmission of an earlier submission. The following is a list of the peer review reports and author responses from that submission.

Round 1

Reviewer 1 Report

Using Landsat data, this study investigated the NDVI trends along the Qinling-Daba Mountains. The authors also disentangle the impacts of climate warming and human activity on the NDVI trends. The methods seem robust but need some clarification. The results and discussion were well-written. This draft does not have a line number, so it is a bit tricky to put comments. I will use paragraph number in each section to provide comments and suggestions.

Major suggestions:

1.     In the methods, please specify the fitting coefficient that you used to harmonize Landsat 8 and 7 dataset.

2.     Please provide how many stations are used in your study area to generate the 1km climate product. If there are few stations, it is worth talking about its related uncertainty in the discussion.

3.     Correlation between climate variable and NDVI trend, why person correlation is used because it only provides marginal correlation.

4.     Please explain why there are striped patches in figure 3 (SLC-off effects from Landsat 7)?

Editorial suggestions:

1.     At the end of the first paragraph, it is worth citing two relevant and recent papers that focused on understanding the impacts of climate change and human activity on vegetation dynamics.

I would write something like: Recent studies have disentangled the impacts of human activity and climate warming on vegetation dynamics at regional (Jiang et al., 2021) and continental scales (Qiu et al., 2020)

Qiu, T. et al., (2020). Urbanization and climate change jointly shift land surface phenology in the northern mid-latitude large cities. Remote Sensing of Environment236, 111477.

Jiang, M. et al., (2021). Disaggregating climatic and anthropogenic influences on vegetation changes in Beijing-Tianjin-Hebei region of China. Science of The Total Environment786, 147574.

2.     End of the second paragraph:

Under the global warming, the in-depth study of vegetation change in this area has great significance to understand climate change.

 Contemporary climate warming requires a more in-depth understanding of vegetation change in this area.

3.     Change “Researches” to “studies” in the first sentence of the third paragraph.

Reviewer 2 Report

Congratulations for this interesting paper,
in the attached text I have indicated some small grammatical corrections, but in general I recommend that you review the text from a linguistic point of view.
Furthermore, I have two observations to make:
1) It would be interesting and useful for the reader to understand which types of vegetation are being examined. In my opinion you could add a brief description of the main types of vegetation present in the area in the introduction to the study area.
2) In the discussions and especially in the conclusions I am not convinced of the statement that there is a positive correlation between human activities and NDVI increase. You should specify in a more convincing way what are the causes that explain the rather unexpected results obtained (the increase of NDVI in the most populated areas). Moreover, It would be interesting, if you have information about it, to pay particular attention to the type of reforestation used and to distinguish the trend of native forests from artificial ones.
